# Household Fuel Use for Heating and Cooking and Respiratory Health in a Low-Income, South African Coastal Community

**DOI:** 10.3390/ijerph16040550

**Published:** 2019-02-14

**Authors:** Sikhumbuzo Archibald Buthelezi, Thandi Kapwata, Bianca Wernecke, Candice Webster, Angela Mathee, Caradee Yael Wright

**Affiliations:** 1Department of Geography, Geoinformatics and Meteorology, University of Pretoria, Pretoria 0002, South Africa; archie@mut.ac.za; 2Mangosuthu University of Technology, Durban 4031, South Africa; 3Environment and Health Research Unit, South African Medical Research Council, Johannesburg 2193, South Africa; thandi.kapwata@mrc.ac.za (T.K.); bianca.wernecke@mrc.ac.za (B.W.); angela.mathee@mrc.ac.za (A.M.); 4University of KwaZulu-Natal, Durban 4041, South Africa; websterc@ukzn.ac.za; 5Environmental Health Department, Faculty of Health Sciences, University of Johannesburg, Johannesburg 2028, South Africa; 6School of Public Health, University of the Witwatersrand, Johannesburg 2000, South Africa; 7Environment and Health Research Unit, South African Medical Research Council, Pretoria 0002, South Africa

**Keywords:** respiratory health, indoor air pollution, household air pollution, South Africa, environmental health, respiratory tract infection

## Abstract

In low-income communities, non-electric fuel sources are typically the main cause of Household Air Pollution (HAP). In Umlazi, a South African coastal, informal settlement, households use electric- and non-electric (coal, wood, gas, paraffin) energy sources for cooking and heating. The study aimed to determine whether respiratory ill health status varied by fuel type use. Using a questionnaire, respondents reported on a range of socio-demographic characteristics, dwelling type, energy use for cooking and heating as well as respiratory health symptoms. Multivariate Poisson regression was used to obtain the adjusted Odds Ratios (ORs) for the effects of electric and non-electric energy sources on prevalence of respiratory infections considering potential confounding factors. Among the 245 households that participated, Upper Respiratory Tract Infections (URTI, *n* = 27) were prevalent in respondents who used non-electric sources compared to electric sources for heating and cooking. There were statistically significant effects of non-electric sources for heating (adjusted OR = 3.6, 95% CI (confidence interval): 1.2–10.1, *p* < 0.05) and cooking (adjusted OR = 2.9, 95% CI: 1.1–7.9, *p* < 0.05) on prevalence of URTIs. There was a statistically significant effect of electric sources for heating (adjusted OR = 2.7, 95% CI: 1.1–6.4, *p* < 0.05) on prevalence of Lower Respiratory Tract Infections (LRTIs) but no evidence for relations between non-electric sources for heating and LRTIs, and electric or non-electric fuel use type for cooking and LRTIs. Energy switching, mixing or stacking could be common in these households that likely made use of multiple energy sources during a typical month depending on access to and availability of electricity, funds to pay for the energy source as well as other socio-economic or cultural factors. The importance of behaviour and social determinants of health in relation to HAP is emphasized.

## 1. Introduction

Human respiratory health is adversely affected by the use of polluting fuels such as coal, wood and paraffin (also known as kerosene) for cooking and indoor space heating [1]. About 4 million people die prematurely each year from diseases attributable to Household Air Pollution (HAP) caused by inefficient cooking practices, paired with solid and liquid fuels, besides poor ventilation practices [1]. HAP has been associated with increased risk of suffering stroke, ischaemic heart disease, chronic obstructive pulmonary disease (COPD) and lung cancer [1,2]. Women and children are especially at risk of exposure to HAP given the amount of time they typically spend indoors. The type of fuel used often determines the nature of the health outcomes, for example, biomass smoke exposure has been associated with an increased risk of chronic bronchitis and COPD, and around 15% of those experiencing long-term exposure to wood smoke suffer from COPD [2,3]. Women exposed to smoke from coal fires also have an elevated risk of lung cancer [2,3]. Similarly, though considered a cleaner fuel than coal or wood, exposure to fumes created by burning paraffin has been shown to impair lung function and increase asthma [4]. 

The burden of disease attributable to HAP, specifically from solid fuel burning, has been estimated for South Africa [5]. In 2000, about 20% of households were exposed to indoor smoke from burning of solid fuels causing 2489 deaths in the country. This figure doubled in the latest Global Burden of Disease study [6] and most of the disease burden was among Black African households. In South Africa, and elsewhere, exposure to HAP from solid fuel burning is a pressing public health problem, particularly in low-income communities [7,8,9,10,11].

In households using wood, paraffin or coal for heating and/or cooking [2], the effects of exposure to indoor air pollution may be exacerbated by other confounding factors including, for example, poor ventilation in the dwelling, inefficient stoves, smoking and burning of incense and mosquito repellent, indoor temperature and relative humidity levels [7,12]. The type of fuel used by low-income communities within and across countries varies depending on, for instance, the resources available or cost of fuel [13]. In Uganda, paraffin is used for lighting and contributes towards HAP in rural communities [14]. Biomass solid fuel, such as cow dung, is preferred in many Indian rural villages for cooking in indoor kitchens [15]. In China on the Tibetan Plateau, wood stoves are a source of heat and contribute greatly to fine particulate matter exposure, especially for women, in rural kitchens [16]. Depending on the source, size and chemical composition of the particulate matter, as well as the duration of the exposure to the suspended particles, different respiratory and cardiovascular health impacts are triggered [17]. Exposure to gaseous indoor pollutants can lead to a variety of health effects by influencing cardiovascular, respiratory and central nervous systems. The physiological status of the exposed person, the pollutant concentration, and the exposure time are all factors which influence the impact of exposure to a certain indoor pollutant on human health [18]. 

In South Africa, low-cost coal is readily available in the interior of the country, near coal mines and coal-fired power stations [19]. In these areas, low-income communities usually use coal for heating and cooking purposes, while electricity is used for lighting. Due to limited availability of wood and coal in certain coastal regions on the eastern seaboard of South Africa, the use of paraffin for heating and cooking is relatively common, including for cooking and heating [20]. Though exposure to pollutants from paraffin is often considered to have a lower health impact relative to solid fuels, fumes from paraffin burning activities have been found to cause Lower Respiratory Tract Infections (LRTIs) such as pneumonia and bronchiolitis [20]. Moreover, health risks are reduced when electric sources of energy, rather than solid or liquid fuels, are used for heating and/or cooking [21]. It is a goal of the South African government to provide electricity to all dwellings in the country [22] with one key benefit being the reduction of exposure to HAP [23]. This study aimed to determine whether respiratory health status in a coastal, low-income community in KwaZulu-Natal differed according to what type of fuel was used (i.e., electric versus non-electric for heating and for cooking). 

## 2. Materials and Methods

### 2.1. Study Area

The study was carried out between 1 April and 31 August 2017 in Umlazi Township in the City of eThekwini, KwaZulu-Natal province, South Africa (Figure 1). Umlazi is the fourth largest township in South Africa covering an area of 47.5 km^2^. It has a population of 404,811 inhabitants living in 104,914 households (on average, four individuals per household). The population is predominantly Black African (99%), and the main language spoken is isiZulu [24].

Umlazi lies within the South Durban Industrial Basin [25] where emissions from oil refineries, chemical plants and heavy traffic contribute further sources of air pollution in the area. Air quality monitoring programmes indicated that the industrial basin has among the highest air pollution levels in the country [26,27].

Two wards from the five located in Umlazi were purposefully selected for inclusion in the study. The two wards had similar socio-economic conditions; however, they differed in respect of housing characteristics. Both wards comprised free-standing and/or semi-detached formal houses (brick, single-story dwellings), and backyard shacks (mixed materials used for walls, single-story) while one of the wards also had hostels (brick, multiple stories) built during the Apartheid era for mass housing of labour. Using a reported asthma prevalence (one of our outcomes of interest) of 10% in children and 12% among adults living in south-central Durban, South Africa [27,28], together with a 5% margin of error and an estimated 80% response rate, the minimum sample size was calculated as 377. Using satellite maps from the eThekwini Town Planning Unit, 400 housing units were randomly selected from the two selected wards.

### 2.2. Survey

A pilot exercise to test and refine the questionnaire was carried out among a sample of participants from both study wards. Thereafter, a cross-sectional study was conducted by randomly selecting two hundred houses (dwellings) from each ward after a random starting point was selected. The random starting point was a corner on a residential block (north-east, north-west, south-east, south-west). A random starting house was selected choosing one of the first three houses on the corner with the help of a list of random numbers [29]. Thereafter, every 10th house was selected for inclusion in the study.

A questionnaire was administered to the head, or primary caregiver, of each selected household, by a trained research assistant. The questionnaire was designed to collect data on demographics, socio-economic status, housing characteristics, fuel use patterns and health outcomes (as shown in Table 1). The questionnaire was made available in both English and isiZulu, and study participants were able to choose which of the two languages they were most comfortable using for the interview. Prior to conducting the interview, participants gave written, informed consent to participate in the study. All data were double entered using EpiData software (version 3.1, EpiData Association, Odensa, Denmark). Permission to conduct the study was obtained from the City of eThekwini and the local ward councillors in advance of the study. Research ethics clearance was granted by the University of Pretoria Research Ethics Committee (Certificate number: GW20160825HS, 3 November 2018).

### 2.3. Questionnaire

The questionnaire (provided in the Appendix A) was based on several models used in epidemiological studies on respiratory diseases, namely the ATS-DLD-78-A questionnaire [30], the Canadian Air Quality and Health Study questionnaire (NHW/HPB-190-03040), the Harvard School of Public Health’s Children’s Health Study Questionnaire (NHW/HPB-190-03210) and the Vaal Triangle Air Pollution and Health Study (VAPS) questionnaire [31]. Personal information collected included gender, age, home language, education level and duration of residence in ward. Respondents were asked about type of dwelling in which they resided, as well as number and purpose of each room.

Respondents were asked about fuel use patterns, including type and frequency of fuel used, for cooking and heating. They were asked about ventilation practices, presence of pets in the house, smoking practices, and presence of mould or mildew on any indoor surfaces. Regarding health, respondents were asked about allergies (hay fever, sinusitis), selected respiratory diseases (bronchitis, pneumonia, asthma) and ill health symptoms (earache, rhinitis, runny nose, wheezing and cough). Further questions aimed to ascertain use of (any) medication, and work absenteeism due to illness.

### 2.4. Data Management and Statistical Analyses

Questionnaire data were coded, captured and underwent quality control checks prior to analysis. Following initial univariate analyses, selected variables were recoded. For example, fuels for cooking and heating were categorized into electric (i.e., electricity) and non-electric (i.e., wood, coal, gas, paraffin) following the practice in a similar study [32].

Respiratory symptoms and illnesses were categorized into Upper (URTIs) and Lower Respiratory Tract Infections (LRTIs). URTI included hay fever, runny nose and ear-ache, and LRTI included wheezing, bronchitis and asthma. The aim was to examine the likelihood of having a URTI or LRTI or not, in relation to exposure to suspected risk (non-electric sources of heating and cooking) and protective factors (electric sources). For this reason, respondents who did not report having any respiratory tract infections were made the reference category allowing for comparison of the magnitude of risk for electric versus non-electric sources. Multivariate Poisson regression was used to estimate the effect of electric and non-electric fuel use (considering confounding factors and co-variates) on URTIs and LRTIs. Risk factors were modelled with associated outcomes, adjusting for other associated risk factors identified from the literature. These factors can be grouped into air pollution sources (e.g., the presence of smokers in a given household or the parental occupation), basic demographics (e.g., employment and educational status of the respondent, as well as hospitalization for respiratory related ailments) and other household-related factors (e.g., housing type and ventilation practices during winter) [32,33].

The associations were expressed as unadjusted and adjusted odds ratios (ORs) along with their 95% confidence intervals (CIs). Missing values were automatically excluded in each model; therefore, each multiple regression model had a different sample size (the smallest being 71 cases of URTIs and LRTIs combined). To obtain adjusted ORs for the effect of using electric or non-electric fuels for heating and cooking on the LRTIs and URTIs, both variables were placed in an initial regression model. This was followed by the addition of a potential confounder in a stepwise manner starting with the most statistically significant from the univariate analysis. At multiple regression level, variables were deemed significant if *p*-values were <0.05. Regression analyses were not conducted for those health outcomes that had fewer than 20 study participants. All statistical analyses were performed using STATA version 14 [34].

## 3. Results

### 3.1. Sample Description and General Profile of the Study Population

A total of 245 households’ respondents completed the questionnaire, resulting in a response rate of 61% (*n* = 245/400). Time constraints to complete the lengthy questionnaire as well as inconvenience for surveys conducted after working hours contributed to the relatively low response rate. More women (61%) than men were respondents to the questionnaire (see Table 1). Most dwellings (80%) were either free-standing or semi-detached structures. Nearly two-thirds of households (62%) relied on electricity for cooking. Paraffin was the second most frequently used fuel for cooking (18%). The predominant energy sources used for heating were non-electric sources when combined, i.e., wood and coal (12%), gas (4%), and paraffin (27%) versus 18% for electric heaters. Twenty-two percent of respondents reported that someone in the home smoked (either cigarettes, cigars and/or a pipe) in the dwelling daily. Most households opened doors and windows in winter months (80%) and pets were not allowed to enter most dwellings. Mould/mildew was found to be present in half of the study dwellings.

### 3.2. Prevalence of Respiratory Health Outcomes

Table 1 lists the levels of respiratory ill-health outcomes, divided into URTIs and LRTIs. Hay fever was the most frequently occurring URTI. Wheezing, a LRTI, was the most commonly reported respiratory ailment. The overall prevalence of LRTIs was higher than that of URTIs.

The prevalence of URTIs and LRTIs was highest among those respondents who used electricity for cooking (although the overall number of URTIs and LRTIs was relatively small) (Table 2). For heating, the prevalence of URTIs was higher among those respondents who reportedly used paraffin. The prevalence of LRTIs among respondents who used electricity versus paraffin was similar (i.e., 30% and 27%, respectively). There were 14 missing responses for energy type used for heating among people who reported with LRTIs. 

### 3.3. Multiple Regression Analysis

Table 3 provides the OR estimates of URTIs and LRTIs by electric versus non-electric energy source used for heating and cooking activities. There were statistically significant effects of non-electric sources for heating and cooking on prevalence of URTIs (although the CIs are relatively wide). There was a statistically significant effect of electric sources for heating on prevalence of LRTIs (again with relatively wide CIs due to the small sample). 

## 4. Discussion

The study aimed to determine whether respiratory ill-health status in a low-income, coastal community in KwaZulu-Natal in which multiple energy sources are used for cooking and heating activities varies according to electric- and non-electric fuel type. Bearing in mind the small sample size, there was some evidence of a relationship between use of non-electric fuels for cooking and heating and URTIs. Also, even in households in which electricity was reportedly used for heating, LRTIs still occurred among individuals living in those homes. It is likely that, despite electricity being the predominant energy source for cooking, other respiratory risk factors (such as non-electric fuels used for heating) exist in the dwelling, the community, as well as in individuals’ occupational settings. While electricity is the most common reported source of energy for cooking, it may not be used all the time and ‘energy switching’ or ‘energy stacking’ may occur, depending on access to and availability of electricity, funds to pay for the energy source as well as other socio-economic or cultural factors like fuel-use preference [35]. Households may make use of one or numerous energy sources at a time to cover their varying energy needs [35]. While this study did not ask specific questions about energy switching, future studies should do so. Electricity, for instance, is commonly used for lighting, refrigeration and entertainment purposes in a low-income community setting, whilst more dirty fuels, such as coal, wood or paraffin are used for heating and cooking. A recent study in Louiville, a low-income community on the Mpumalanga Highveld in South Africa, showed that households rely on wood and paraffin, but also electricity, to complete their basic household activities [35]. Although up to 70% of households in Louiville were connected to the electricity grid, the high price of electricity limited its usage, and households resorted to using cheaper, more dirty fuel types (namely wood, coal and paraffin) for activities such as cooking and heating [36]. In Samora Machel, a low-income community in Cape Town, people relied mostly on paraffin and wood for their main energy needs, and 38% of households used electricity for radio, refrigeration and communication activities [37].

Though the results of this study show that electricity represents a main energy source for cooking, the wide range of energy sources listed by households for cooking and heating indicates that the probability of energy mixing/stacking at a household level in these homes in Umlazi is likely to be high. The South African 2011 Census states that between 70% and 90% of surveyed households made use of electricity for heating, cooking and lighting and that less than 10% of households used paraffin or other dirty fuels for the same activities [24]. In Umlazi, in which almost a quarter of the people have no source of income [24] it is very likely that the concept of energy poverty, i.e., the lack of access to modern energy services necessary for human development [38] is a reality and electricity may not be used all-month long.

Half of the respondents reported mildew and mould in their homes. Damp and mouldy housing have been linked to chronic LRTIs [39,40]. Given that relative humidity is typically high in Umlazi, since it is a coastal town in a subtropical area, it is reasonable to state that the mould could represent an important risk factor contributing to LRTIs in Umlazi. Households in which electricity is most commonly used for heating and cooking purposes were reported to have presence of mould, and so those suffering from respiratory problems in these households were possibly suffering from respiratory impacts resulting from fungal aerosol exposure.

One in five respondents indicated the presence of ETS in their homes. Exposure to active and passive smoking is associated with increased risk of respiratory infections [41,42]. In those households in which smokers were present, respiratory infections may have been exacerbated by smoke inhalation, either active or passive. A similar exposure-response study in Kenya showed that smoking increased the risk of URTIs in a low-income community setting in which indoor air pollution was prevalent [43].

Eight out of ten households reportedly ventilated their homes even during winter months. Increased ventilation typically decreases the concentration of indoor air pollution due to mixing and dilution of indoor air though the infiltration of ambient air into the home and through the escape of indoor air pollution into the ambient environment [44]. In wintertime, however, it is well documented in many South African low-income communities that ambient air quality present outdoors is particularly poor [45], and infiltration of pollution into households, which use mainly electricity for heating and cooking purposes, is a possible contributing factor to indoor air pollution. As Umlazi is heavily impacted by the industrial activities taking place in the so-called “pollution hot spot” of South Durban Basin, it is likely that infiltration of outdoor air pollution into homes occurred [46,47,48]. 

The study sample size was constrained by budget and resources; however, a representative sample was achieved by random starting point selection in at two wards in Umlazi. A low-response rate is not uncommon in cross-sectional household surveys conducted in urban townships [49], however, we did attempt multiple house visits by fieldworkers during recruitment to increase sample size. Despite questionnaire piloting and pre-site field visits, we did not consider the possibility of fuel switching occurring in Umlazi and additional questions should have been included on this topic in the study questionnaire. Small changes to some questions’ structure, for example, separation of wood and coal as two distinct solid fuel types, would have been advantageous too. The questionnaire should also have asked about lighting. Indoor air quality exposure and measurement of indoor temperatures within households would have given additional insight into HAP exposure, however, this was not possible within the constraints of the study scope.

Due to the relatively small sample size and the low use of wood/coal (1% of households used wood/coal for cooking/heating) no associations were found between URTIs or LRTIs and wood/coal for cooking or heating purposes. There is sound evidence of the relationship between indoor burning of wood/coal indoors and respiratory morbidity and mortality, including in South African [50,51]. Electricity is the recommended energy source for indoor use in homes to reduce HAP according to the World Health Organization [52]. In our study, it is likely that the expense and perhaps even the availability of electricity (‘loadshedding’ or planned power outages are common in South Africa) constrained its continuous use, and these factors should be further investigated in Umlazi.

## 5. Conclusions

Despite the small sample size, we found some differences in respiratory health status in a coastal, low-income community in KwaZulu-Natal according to type of fuel (electric versus non-electric for heating and for cooking). Electricity was the main fuel source used for cooking and non-electric sources (combined paraffin, wood, coal and gas) for heating. There were statistically significant effects of non-electric sources for heating and cooking on prevalence of URTIs, and surprisingly a statistically significant effect of electric sources used for heating on prevalence of LRTIs (although CIs were relatively wide). For the latter, we consider that energy switching could be common in these households that likely made use of multiple energy sources during a month. The importance of behaviour and socio-economic determinants in relation to HAP is critical when defining a comprehensive approach to understanding the relationship between respiratory health status, air pollution and fuel use type.

## Figures and Tables

**Figure 1 ijerph-16-00550-f001:**
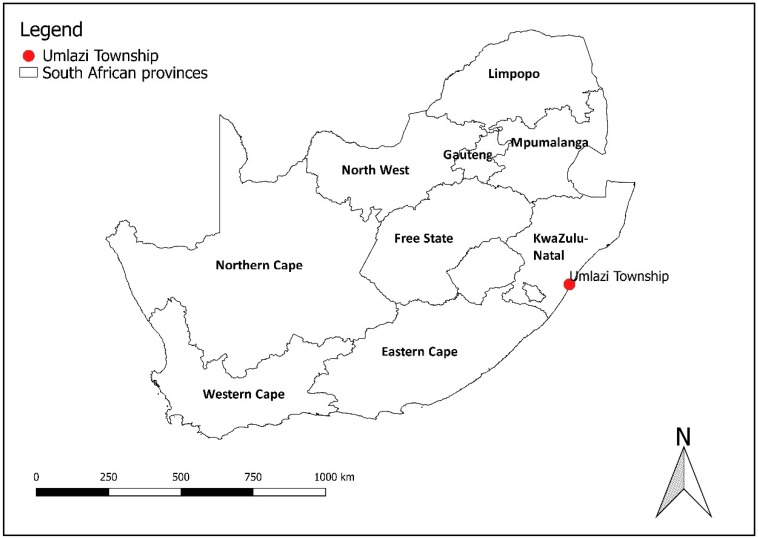
Umlazi Township in relation to the province of KwaZulu Natal, South Africa.

**Table 1 ijerph-16-00550-t001:** Demographic variables and living conditions of the study population (*n* = 245).

Questionnaire Variable	Response Item	Percentage of Participants *n* = 245 (%)
Gender of the respondent	Male	38.9
	Female	60.7
	Missing	0.4
Type of home	Single unattached dwelling	44.9
	Single attached dwelling	39.4
	Hostel	14.5
	Missing	2.0
Main fuel used mostly for cooking	Electricity	61.6
	Paraffin	17.7
	Gas	13.9
	Electricity and Gas	5.5
	Wood/coal	1.2
	Missing	3.2
All heating methods used in the household	Portable electric heater	17.6
	Gas heater	4.1
	Paraffin	27.3
	Wood/coal	11.8
	Missing	38.8
Open doors and windows in winter	No	19.7
	Yes	80.3
	Missing	0.4
Mould or mildew in the home	No	50.2
	Yes	49.8
	Missing	0.0
Pets allowed in the home	No	92.2
	Yes	7.8
	Missing	0.0
A household member smokes inside the dwelling on a daily basis	No	77.5
	Yes	21.6
	Missing	0.82
Does the respondent currently smoke?	No	78.8
	Yes	21.2
Health status—URTI (past month)	Hay fever	7.8
	Runny nose	3.7
	Ear ache	2.5
Health status—LRTI (past year)	Wheezing	15.9
	Bronchitis	6.9
	Asthma	3.7

**Table 2 ijerph-16-00550-t002:** Prevalence of grouped URTIs and LRTIs according to individual energy source for heating and cooking.

Energy Source		URTI *n* = 27 *n* (%)	LRTI *n* = 44 *n* (%)
Used for cooking	Electricity	15 (57.7)	25 (62.5)
	Paraffin	4 (15.4)	9 (22.50)
	Gas	6 (23.1)	2 (5.0)
	Wood/coal	0 (0.0)	0 (0.0)
	Missing	2	8
Used for heating	Portable electric heater	6 (22.2)	13 (29.6)
	Gas heater	2 (7.4)	4 (9.1)
	Paraffin	13 (48.2)	12 (27.3)
	Wood/coal	0 (0.0)	1 (2.3)
	Missing	6	14

URTI: Upper Respiratory Tract Infection; LRTI: Lower Respiratory Tract Infection.

**Table 3 ijerph-16-00550-t003:** Odds ratio estimates of effects of electric versus non-electric energy sources for heating and cooking on prevalence or URTIs and LRTIs *.

Energy Source	Prevalence (%)	*p*-Value	OR (95% CI)	Adjusted *p*-Value	Adjusted OR (95% CI) ^a^
**Any URTI (*n* = 27 **)**					
Heating					
Non-electric ^a^	33.3	<0.05	3.0 (1.3–6.7)	<0.05	3.6 (1.2–10.1)
Electric ^b^	22.2	0.50	1.4 (0.5–3.7)	0.90	1.1 (0.3–4.4)
Cooking					
Non-electric ^c^	37.0	0.08	2.1 (0.9–4.9)	<0.05	2.9 (1.1–7.9)
Electric ^d^	18.5	0.65	0.8 (0.3–1.8)	0.37	0.6 (0.2–1.7)
**Any LRTI (*n* = 44 **)**					
Heating					
Non-electric ^e^	43.2	0.32	1.4 (0.2–2.7)	0.28	1.5 (0.7–3.3)
Electric ^f^	29.5	<0.05	2.3 (1.1–5.0)	<0.05	2.7 (1.1–6.4)
Cooking					
Non-electric ^g^	25.0	0.76	1.1 (0.5–2.4)	0.95	1.0 (0.3–3.0)
Electric ^h^	56.8	0.68	0.8 (0.4–1.6)	0.72	0.8 (0.3–2.0)

* Respondents who did not report any LRTI and URTI were used as the reference category. ** Numbers of URTIs and LRTIs by fuel type do not add up to 27 and 44 (i.e., to 100%), respectively, due to missing data. ^a^ Model was adjusted for current smoking status, employment status, working in a dusty environment, leaving windows and doors open in winter, Environmental Tobacco Smoke (ETS) exposure at home. ^b^ Model adjusted for current smoking status, employment status, leaving windows and doors open in winter, ETS exposure at home, previous hospitalization for respiratory related ailments, house type. ^c^ Model was adjusted for current smoking status, level of education, employment status. ^d^ Model was adjusted for employment status, previous hospitalization for respiratory related ailments, working in a dusty environment, leaving windows and doors open in winter, current smoking status. ^e^ Model adjusted for current smoking status, employment status, working in a dusty environment. ^f^ Model was adjusted for current smoking status, employment status, working in a dusty environment, leaving windows and doors open in winter, ETS exposure at home. ^g^ Model was adjusted for current smoking status, level of education, employment status. ^h^ Model was adjusted for level of education, employment status, current smoking status.

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
