# Peer review of "Household Fuel Use for Heating and Cooking and Respiratory Health in a Low-Income, South African Coastal Community"

_ijerph, 2019, doi:10.3390/ijerph16040550_

Round 1

Reviewer 1 Report

Brief summary

The objective of the research was to determine the association between the prevalence of respiratory symptoms categorized in low respiratory tract infections (LRTI) and upper respiratory tract infections (URTI). The prevalence of both were estimated as was reported by a household survey and the application of a questionnaire. Prevalence of suffering LRTI and URTI were associated with the use of two categories of fuels; clean fuel (electricity) and polluting fuels (paraffin, coal, biomass, LPG). The main contribution of the article is that it approaches the qualitative state of indoor air pollution in South Africa, as well as its impact on respiratory diseases. Nevertheless, the research shows statistical results that are based on small sample size (71). Due to this, few relationships were statistically significant and the expected effects were discordant. For example, the use of electricity for heating was associated with lower development of URTI but with a higher development of LRTI. 

Broad comments:

1) Abstract: can be improved in the following points:

- indicate the sample size of the total households surveyed and used for the regression analysis.

- indicate in methods the type of regression used.

- in results indicate the relationships that resulted with a significant association, as well as those that did not.

2) Introduction: it has an adequate extension, however, it can be improved as indicated in the specific comments. 

- Also, I suggest giving mention to indoor air pollutants that affect human health, such as particulate matter and carbon monoxide. 

- The estimated mortality data in South Africa, based on reference 5, can be complemented with the number indicated in the last Global Burden of Disease (GBD) study (2017) for this country (https://doi.org/10.1016/S0140-6736(18) 32225-6). In the GBD webpage, there is a tool to visualize the number of deaths disaggregated by country and risk (http://ghdx.healthdata.org/gbd-results-tool). Please considerate this research as the estimated number increases to about 4,000 deaths in 2017. 

- References 13, 14 and 15 seem to be interchanged and do not correspond to what is stated in the sentence, please verify the correct reference.

3) Methods and Results: please consider what is suggested in Specific Comments.

4) Discussion and Conclusion: 

The discussion is somewhat extensive. It is suggested to pass the first three paragraphs to the Introduction.

The possible relationship between the increase in LRTI due to the use of electric heating should be discussed in greater depth considering that the estimated relationships are based on low sample size.

In Conclusion, the total sample number should be clarified, including only the answers for URTI and LRTI after removing missing answers.

Specific comments:

1) Abstract:

- in Background, on line 20, I suggest changing the phrase that begins with "it was assumed that households ..." by "households use electric and non-electric sources (paraffin, coal, biomass) for domestic duties as cooking and heating".

- in Result, line 26, I suggest eliminating the first two sentences because they are little informative and deviate the meaning of the article, which is the statistical relationship of the respiratory symptoms with the fuel used in households. 

- in Conclusion

2) Introduction: my editing suggestions are as follows.

- on line 41, change to the phrase "caused by inefficient combustion devices paired with solid and liquid fuels, besides poor ventilation".

- on line 42, "HAP has been associated with increased risk of suffering stroke, ...".  The reference shown here could be complemented with the reference indicated with the number 10 (Ezzati et al., 2005).

- In the paragraph beginning at line 57, change to "In households using wood .. by other factors ...".

- line 59 change "poor quality" to "inefficient". Change "indoor smoking activities" to "smoking and burning of incense and mosquito repellent".

- line 66, ending the paragraph after reference 15. Begin the following with the sentence introducing the scenario in South Africa and add the text shown on line 69. Change "and electricity for lighting" to "while electricity is used for lighting".

- line 69 discard "around eThekwini". The suggested phrase is "... coastal regions on the eastern seaboard of South Africa, ..."

- line 75 change to "A goal of the South African government is to provide ...". Please indicate in this sentence if the target is really "all" the dwellings or if they have a specific goal (percentage, etc.).

- Line 77, change "ascertain" to determine, assess or evaluate. Change "... in a low-income community in KwaZulu-Natal ..." to "... in a coastal low-income community differed according to the type of fuel ..."

3) Methods

- Change the subtitle of "Sample" by "Study Area".

- Write the dates in English format (April 1st and August 31st, 2017).

- line 83 change to "Umlazi is the fourth largest township in South Africa, covering an area of ...".

- line 84 change "people" for "inhabitants".

- line 85 discard "this amount to" and leave "on average, this represents four individuals per household".

- line 94 delete "of range".

- Delete sentence written in lines 95 and 97, replace by a sentence that mentions physical characteristics of the dwellings (material, 1 or 2 floors, etc.).

- The sentence started on line 97, "based on asthma prevalence ..." is not clear. Please indicate only the number of pre-selected houses and the procedure (random number generation, random hand selection, etc.).

- Change the subtitle 2.2 "Procedures" to "Survey".

- Start text on line 103 with "A cross-sectional survey was conducted ..."

- line 111, change "see below" to "as shown in Table ..."

- line 121 change the first sentence to "The questionnaire was based on several models used in ..."

- line 126 discard "date of birth" and leave only "age".

- line 132, it is mentioned that the questionnaire inquired about respiratory symptoms, please include in supplementary material the questionnaire used or an extract of the questions used to consult about the health status. Reference in this paragraph to supplementary material.

4) Results

- line 172, indicate in parentheses the percentage of use in wood, coal, LPG, paraffin. Discard the sentence about 1/3 of households using paraffin.

- line 182, exchange the order of sentences. Leave the URTI sentence first, since in Table 2 it appears first.

5) Discussion and Conclusion: please refer to Broad Comments.

Author Response

Brief summary

The objective of the research was to determine the association between the prevalence of respiratory symptoms categorized in low respiratory tract infections (LRTI) and upper respiratory tract infections (URTI). The prevalence of both were estimated as was reported by a household survey and the application of a questionnaire. Prevalence of suffering LRTI and URTI were associated with the use of two categories of fuels; clean fuel (electricity) and polluting fuels (paraffin, coal, biomass, LPG). The main contribution of the article is that it approaches the qualitative state of indoor air pollution in South Africa, as well as its impact on respiratory diseases. Nevertheless, the research shows statistical results that are based on small sample size (71). Due to this, few relationships were statistically significant, and the expected effects were discordant. For example, the use of electricity for heating was associated with lower development of URTI but with a higher development of LRTI. 

>>> We are aware of the challenges of the small sample size and tried to make this as clear as possible as a study limitation in the manuscript. We have also made the findings of the study clearer as best we could and checked for consistency of what is reported throughout the manuscript.

Broad comments:

1) Abstract: can be improved in the following points:

- indicate the sample size of the total households surveyed and used for the regression analysis.

>>> We have indicated the sample size of households surveyed as 245 and 71 (URTI = 27 and LRTI = 44) in the regression analysis now in the abstract.

- indicate in methods the type of regression used.

>>> We used multivariate Poisson regression to obtain adjusted OR for the effects of electric and non-electric sources of heating on prevalence of URTIs and LRTIs in the study and we have indicated this in the abstract.

- in results indicate the relationships that resulted with a significant association, as well as those that did not.

>>> Owing to word requirements for the IJERPH abstract, we have included our most significant association in the abstract and added the ones that were not significant in the abstract.

2) Introduction: it has an adequate extension; however, it can be improved as indicated in the specific comments. 

- Also, I suggest giving mention to indoor air pollutants that affect human health, such as particulate matter and carbon monoxide. 

>>>We have added the following text to the introduction: Depending on the source, size and chemical composition of the particulate matter, as well as the duration of the exposure to the suspended particles, different respiratory and cardiovascular health impacts are triggered [16]. Similarly, exposure to gaseous indoor pollutants can lead to a variety of health effects by influencing cardiovascular, respiratory and central nervous systems. The physiological status of the exposed person, the pollutant concentration, and the exposure time are all factors which influence how much the exposure to a certain indoor pollutant impacts on human health [17].

- The estimated mortality data in South Africa, based on reference 5, can be complemented with the number indicated in the last Global Burden of Disease (GBD) study (2017) for this country (https://doi.org/10.1016/S0140-6736(18) 32225-6). In the GBD webpage, there is a tool to visualize the number of deaths disaggregated by country and risk (http://ghdx.healthdata.org/gbd-results-tool). Please considerate this research as the estimated number increases to about 4,000 deaths in 2017. 

>>>We thank the reviewer for this suggestion. We have included the Lancet reference in our introduction as the Reviewer suggested.

- References 13, 14 and 15 seem to be interchanged and do not correspond to what is stated in the sentence, please verify the correct reference.

>>>Thank you for noticing this. We have fixed the references.

3) Methods and Results: please consider what is suggested in Specific Comments.

>>> We thank the Reviewer for their comprehensive comments which we have done our best to attend to and improve the manuscript.

4) Discussion and Conclusion: 

The discussion is somewhat extensive. It is suggested to pass the first three paragraphs to the Introduction.

>>>We have made an extensive effort to tighten up the discussion, however, we prefer to retain the discussion regarding energy switching here since it was a finding from our study. We see evidence of energy switching in our data, and we were not aware of this at the start of the study.

The possible relationship between the increase in LRTI due to the use of electric heating should be discussed in greater depth considering that the estimated relationships are based on low sample size.

>>>We have done so by adding additional discussion in the limitations section that these results were based on a relatively small sample size and that further studies in similar settings are required to corroborate these findings.

In Conclusion, the total sample number should be clarified, including only the answers for URTI and LRTI after removing missing answers.

>>>We have noted the small sample size in the conclusion as the Reviewer has advised us to do.

Specific comments:

1) Abstract:

- in Background, on line 20, I suggest changing the phrase that begins with "it was assumed that households ..." by "households use electric and non-electric sources (paraffin, coal, biomass) for domestic duties as cooking and heating".

>>>We have made this amendment.

- in Result, line 26, I suggest eliminating the first two sentences because they are little informative and deviate the meaning of the article, which is the statistical relationship of the respiratory symptoms with the fuel used in households. 

>>>We have removed these two sentences.

- in Conclusion

>>> We are not sure if the reviewer meant to write something here.

2) Introduction: my editing suggestions are as follows.

- on line 41, change to the phrase "caused by inefficient combustion devices paired with solid and liquid fuels, besides poor ventilation".

>>>Thank you. We have made this amendment.

- on line 42, "HAP has been associated with increased risk of suffering stroke, ...".  The reference shown here could be complemented with the reference indicated with the number 10 (Ezzati et al., 2005).

>>>We have made the amendment to the sentence and we have included the Ezzati reference here too.

- In the paragraph beginning at line 57, change to "In households using wood .. by other factors ...".

>>>We have made this amendment.

- line 59 change "poor quality" to "inefficient". Change "indoor smoking activities" to "smoking and burning of incense and mosquito repellent".

>>>We have made this amendment.

- line 66, ending the paragraph after reference 15. Begin the following with the sentence introducing the scenario in South Africa and add the text shown on line 69. Change "and electricity for lighting" to "while electricity is used for lighting".

>>>We have made these amendments.

- line 69 discard "around eThekwini". The suggested phrase is "... coastal regions on the eastern seaboard of South Africa, ..."

>>>We have deleted this text.

- line 75 change to "A goal of the South African government is to provide ...". Please indicate in this sentence if the target is really "all" the dwellings or if they have a specific goal (percentage, etc.).

>>>The goal of the government is to all dwellings and we have clarified this in the text.

- Line 77, change "ascertain" to determine, assess or evaluate. Change "... in a low-income community in KwaZulu-Natal ..." to "... in a coastal low-income community differed according to the type of fuel ..."

>>>Thank you. We have made these amendments.

3) Methods

- Change the subtitle of "Sample" by "Study Area".

>>> We have made this amendment.

- Write the dates in English format (April 1st and August 31st, 2017).

>>>We have amended the dates to American formatting. We had been using British date formatting.

- line 83 change to "Umlazi is the fourth largest township in South Africa, covering an area of ...".

>>>We have made this amendment.

- line 84 change "people" for "inhabitants".

>>>We have made this amendment.

- line 85 discard "this amount to" and leave "on average, this represents four individuals per household".

>>>We have made this amendment.

- line 94 delete "of range".

>>>We have made this amendment.

- Delete sentence written in lines 95 and 97, replace by a sentence that mentions physical characteristics of the dwellings (material, 1 or 2 floors, etc.).

>>> We have amended this sentence as follows: “Both wards comprised free-standing and/or semi-detached formal houses (brick, single-story dwellings), and backyard shacks (mixed materials used for walls, single-story) while one of the wards (Ward 76) also had hostels (brick, multiple stories), built during the Apartheid era for mass housing of labour.”

- The sentence started on line 97, "based on asthma prevalence ..." is not clear. Please indicate only the number of pre-selected houses and the procedure (random number generation, random hand selection, etc.).

>>> We have rephrased the sentence and hope that this conveys the points in a better way. We would like to retain the sample size calculation information since Readers often request this information from us. The test reads as follows: “Using a reported asthma prevalence (one of our outcomes of interest) of 10% in children and 12% among adults living in south-central Durban, South Africa [24,25], together with a 5% margin of error and an estimated 80% response rate, the minimum sample size was calculated as 377. Using satellite maps from the eThekwini Town Planning Unit, 400 housing units were randomly selected from the pre-determined study area (i.e. from the two selected wards).”

- Change the subtitle 2.2 "Procedures" to "Survey".

>>>We have made this amendment.

- Start text on line 103 with "A cross-sectional survey was conducted ..."

>>>Reviewer 2 requested more information about the pilot study which we have added as the first sentence to this paragraph. The second sentence was amended as follows: “Thereafter, a cross-sectional study was conducted in the selected wards in Umlazi Township.”

- line 111, change "see below" to "as shown in Table ..."

>>>We have made this amendment.

- line 121 change the first sentence to "The questionnaire was based on several models used in ..."

>>>We have made this amendment.

- line 126 discard "date of birth" and leave only "age".

>>>We have made this amendment.

- line 132, it is mentioned that the questionnaire inquired about respiratory symptoms, please include in supplementary material the questionnaire used or an extract of the questions used to consult about the health status. Reference in this paragraph to supplementary material.

>>>We have included the full questionnaire as supplementary material.

4) Results

- line 172, indicate in parentheses the percentage of use in wood, coal, LPG, paraffin. Discard the sentence about 1/3 of households using paraffin.

>>>We have added the percentages and removed the paraffin phrase.

- line 182, exchange the order of sentences. Leave the URTI sentence first, since in Table 2 it appears first.

>>>We have made this amendment.

5) Discussion and Conclusion: please refer to Broad Comments.

>>> We thank the Reviewer for providing us with so many comments which have helped us including the PhD candidate to improve the manuscript.

Reviewer 2 Report

This study examined the respiratory health effects of indoor air pollution from household fuel combustion for heating and cooking a low-income, coastal community. This is an important topic with great importance in public health. I provided a few comments for the authors to consider:

1.     Please give more details about the pilot study.

2.     The statistical methods should be a mix-effect model, as they assessed the people living in the same ward/family may share some similar characteristics. Please refer to a few similar studies: Ritz B, (2007). American Journal of Epidemiology 166(9): 1045-1052. Lin H, (2017). Hypertension 69(5): 806-812.

3.     The classification can be modified, for example, gas is kind of clean fuel in terms of air pollution emission.

4.     Are there any difference between the included and excluded subjects?

5.     In Table 3, both groups (Non-electric and Electric) had an OR, so what is the reference group?

6.     I wonder whether the local residents had the habit to burn incense indoors, as this is also an important air pollution source, and has been associated with various health outcomes, for example: Song X, (2017). The Association of Domestic Incense Burning with Hypertension and Blood Pressure in Guangdong, China. International Journal of Environmental Research and Public Health 14(7): 788.

Author Response

This study examined the respiratory health effects of indoor air pollution from household fuel combustion for heating and cooking a low-income, coastal community. This is an important topic with great importance in public health.

>>> We thank the Reviewer for their comments.

I provided a few comments for the authors to consider:

1.        Please give more details about the pilot study.

>>>We have added the following information about the pilot study: “A pilot exercise to test and refine the questionnaire was carried out among a purposefully selected sample of 10 participants from both of the two study wards included in the study.”

2.        The statistical methods should be a mix-effect model, as they assessed the people living in the same ward/family may share some similar characteristics. Please refer to a few similar studies: Ritz B, (2007). American Journal of Epidemiology 166(9): 1045-1052. Lin H, (2017). Hypertension 69(5): 806-812.

>>>We thank the Reviewer for clarifying this and we have now included mention of the method we used, namely, multivariate Poisson regression due to our small sample size, in the manuscript. The studies that the Reviewer has guided us to read have very large sample sizes and therefore the mixed-effect method was appropriate in their studies.

3.        The classification can be modified, for example, gas is kind of clean fuel in terms of air pollution emission.

>>> We agree with the Reviewer that we could have placed gas in a different category since it is a cleaner fuel than coal and wood, however, we have chosen to simplify the categorization with electric version non-electric options. This is also since our government plans to roll out electricity to all South Africans homes.

4.        Are there any difference between the included and excluded subjects?

>>> No, there were no differences between included and excluded participants who declined to participate. All participants came from similar suburbs.

5.        In Table 3, both groups (Non-electric and Electric) had an OR, so what is the reference group?

>>> Respondents who did not report any LRTI and URTI were used as the reference category – and this is stated as a note beneath Table 3.

6.        I wonder whether the local residents had the habit to burn incense indoors, as this is also an important air pollution source, and has been associated with various health outcomes, for example: Song X, (2017). The Association of Domestic Incense Burning with Hypertension and Blood Pressure in Guangdong, China. International Journal of Environmental Research and Public Health 14(7): 788.

>>> No questions were asked regarding burning of incense in the questionnaire. In terms of Nguni culture, burning of incense is a confidential practice of connecting with ancestors. Except in a training camp for traditional healers, it is not common practice to burn incense since it is used only during cultural occasions in most Nguni cultures by the elders.